https://doi.org/10.1038/s41467-020-15977-4　　**OPEN**

# Model-driven generation of artificial yeast promoters

Benjamin J. Kotopka [1] & Christina D. Smolke [1,2]✉

Promoters play a central role in controlling gene regulation; however, a small set of promoters is used for most genetic construct design in the yeast *Saccharomyces cerevisiae*. Generating and utilizing models that accurately predict protein expression from promoter sequences would enable rapid generation of useful promoters and facilitate synthetic biology efforts in this model organism. We measure the gene expression activity of over 675,000 sequences in a constitutive promoter library and over 327,000 sequences in an inducible promoter library. Training an ensemble of convolutional neural networks jointly on the two data sets enables very high ($R^2 > 0.79$) predictive accuracies on multiple sequence-activity prediction tasks. We describe model-guided design strategies that yield large, sequence-diverse sets of promoters exhibiting activities higher than those represented in training data and similar to current best-in-class sequences. Our results show the value of model-guided design as an approach for generating useful DNA parts.

[1] Department of Bioengineering, Stanford University, Stanford, CA 94305, USA. [2] Chan Zuckerberg Biohub, San Francisco, CA 94158, USA.
✉email: csmolke@stanford.edu

Promoters are critical in regulating protein expression, a key task for biological systems. In bioengineered systems, precise gene expression control is needed for balancing enzyme expression levels in engineered metabolic pathways[1–3] and building gene circuits to control cell behavior[4,5]. Thus, access to large promoter sets with useful properties may advance genetic construct design.

Construct design in the model yeast *Saccharomyces cerevisiae* currently relies on a small number of native promoters. While additional useful sequences have been uncovered by genome mining[6], natural genomes contain a limited number of strong promoters that reliably induce high levels of protein production. Methods for generating artificial promoters offer an alternative source of useful promoters.

Typically, sequence-diverse[2], short[7], and transcriptionally active sequences are desired. Artificial promoter libraries have been generated through mutagenesis of a wild-type template[8] or by screening libraries[7,9]. Alternatively, natural promoters can be rationally modified; for example, binding sites for the artificial transcription factor ZEV, which induces expression in the presence of beta-estradiol, were introduced into the yeast $P_{GAL1}$ and $P_{CYC1}$ promoters to create a set of inducible sequences[10]. However, mutagenesis-derived promoters have similar sequences, which may complicate the use of homology-guided sequence assembly methods[11], and assembly from random sequences can require screening tens of millions of constructs to identify a handful of useful promoters[7].

In contrast, model-guided approaches to sequence design can deliver sequences exhibiting desirable biological properties. In one example, a Hidden Markov Model predicting nucleosome occupancy from sequence[12] using yeast nucleosome positioning data[13] was used to boost expression in both native and artificial promoters[14]. Such approaches require large data sets to train an accurate model.

Massively parallel reporter assays (MPRAs), which characterize large ($10^5$–$10^8$) DNA sequence libraries, can provide the needed data. FACS-seq is an MPRA[15,16] for measuring gene-regulatory activities in a single fluorescence-activated cell sorting (FACS) sort and next-generation sequencing experiment[15,16]. In this technique, a library is sorted by the abundance of a fluorescent protein regulated by the sequence of interest. The distribution of cells in each sort bin is used to estimate protein production. FACS-seq has been used to characterize libraries of randomized 5′ untranslated regions (UTRs)[17] and short, complete artificial promoters[18–20] in yeast. These data sets were used to investigate the effect on promoter activity of hand-selected sequence properties[17–20].

Additionally, researchers have built models to predict promoter activity directly from sequence. Deep learning techniques such as convolutional neural networks (CNNs) have performed well on genomics modeling tasks[21,22]. CNNs were applied to modeling yeast MPRA data sets[23], and CNNs trained on MPRA data sets of artificial 5′ UTRs were exploited to design 5′ UTRs in both yeast and human cells[23,24]. However, exploiting this approach for synthetic biology applications requires modeling longer and more complex full-scale promoters, making data collection and modeling more challenging.

To model and design entire promoters, we adopt a strategy that borrows conserved motifs from known promoters and seeks to learn a sequence−function relationship for the spacer sequences between these motifs. We perform FACS-seq on two libraries comprising over 675,000 constitutive and over 327,000 ZEV-inducible promoters. Using these data sets, we develop highly accurate predictive models of promoter activity. We implement model-guided sequence design strategies to generate large, sequence-diverse promoter sets, which we confirm to be highly

active in vivo. In silico mutagenesis of designed sequences elucidates sequence features identified as significant by the model. Our work provides promoter sets with useful properties for synthetic biology applications, as well as a tool for generating promoters with useful properties, demonstrating the value of CNNs trained on MPRA-generated data for designing complex functional DNA sequences.

## Results

**High-throughput promoter library characterization**. We developed a promoter activity model in order to generate useful promoters. We first characterized a promoter library based on $P_{GPD}$ (also called $P_{TDH3}$). $P_{GPD}$ reliably exhibits high activity[25], making it popular in promoter engineering[26].

Conserved motifs in yeast promoters include transcription factor binding sites (TFBSes) and general transcription factor motifs, such as the TATA box[27]. While the transcription start site (TSS) can vary[28], certain motifs are preferred[29,30]. The less-conserved spacer sequences between these motifs also influence activity, particularly the core promoter from the TATA box to the TSS[31] and the 5′ UTR[32]. We identified conserved motifs in $P_{GPD}$, including TFBSes for Rap1p and Gcr1p, the TATA box, and the TSS through published literature[33] and the JASPAR database[34]. We defined $P_{GPD}$ as starting approximately 100 bp upstream of the first Rap1p site. This sequence was used to design libraries (Supplementary Fig. 1).

Promoter libraries were designed to preserve conserved motifs and randomize spacer sequences. Most spacers were designed with equal frequencies of the four bases. However, because the abundance of T nucleotides in the core promoter strongly affects activity[31], we designed this region with base frequencies matching the original $P_{GPD}$ core promoter. G was excluded after the TSS to avoid premature ATG start codons, which weaken activity[17].

We designed seven $P_{GPD}$ libraries to test four parameters (Supplementary Fig. 2): length of Rap1p motifs, base frequency in spacers between the TFBSes, spacer length between the TFBSes and the TATA motif, and TATA motif length. The libraries were generated by PCR-amplifying oligonucleotides containing the constant and randomized regions. Amplification products were Golden Gate-assembled and cloned into a two-color plasmid, in which green fluorescent protein (GFP) expression was driven by the promoters, and mCherry expression was driven by constitutive $P_{TEF1}$ (Fig. 1a). To control for expression noise, the GFP: mCherry fluorescence ratio served as an expression activity measure[35] (Supplementary Fig. 3).

We determined the expression activity distribution in the libraries via flow cytometry after transforming them into yeast strain CSY3 (W303 MATα) (Supplementary Fig. 2). Only deleting the Rap1p binding sites had a marked impact on the activity distribution. We designed a final library (Final, Supplementary Fig. 2) to maximize library diversity without sacrificing activity. This library included 10-bp Rap1p sites, short constant regions, and long randomized spacers. The expression activities spanned over an order of magnitude, with the highest activities approximately three times lower than $P_{GPD}$ (Supplementary Fig. 2). Sequences in this library were 312 bp long, with 83% of the sequence randomized (Supplementary Fig. 4).

We designed inducible promoter libraries based on the ZEV system[36]. We characterized four $P_{ZEV}$ libraries, varying the number of ATF binding sites, internal spacer length, and TATA motif length (Supplementary Fig. 5). We generated a strain expressing the ZEV ATF from $P_{ACT1}$ (CSY1252) and characterized activity in the presence of 0, 0.01, and 1 μM beta-estradiol. We observed similar uninduced activities for all designs. Activities in 1 μM beta-estradiol were higher in designs with the

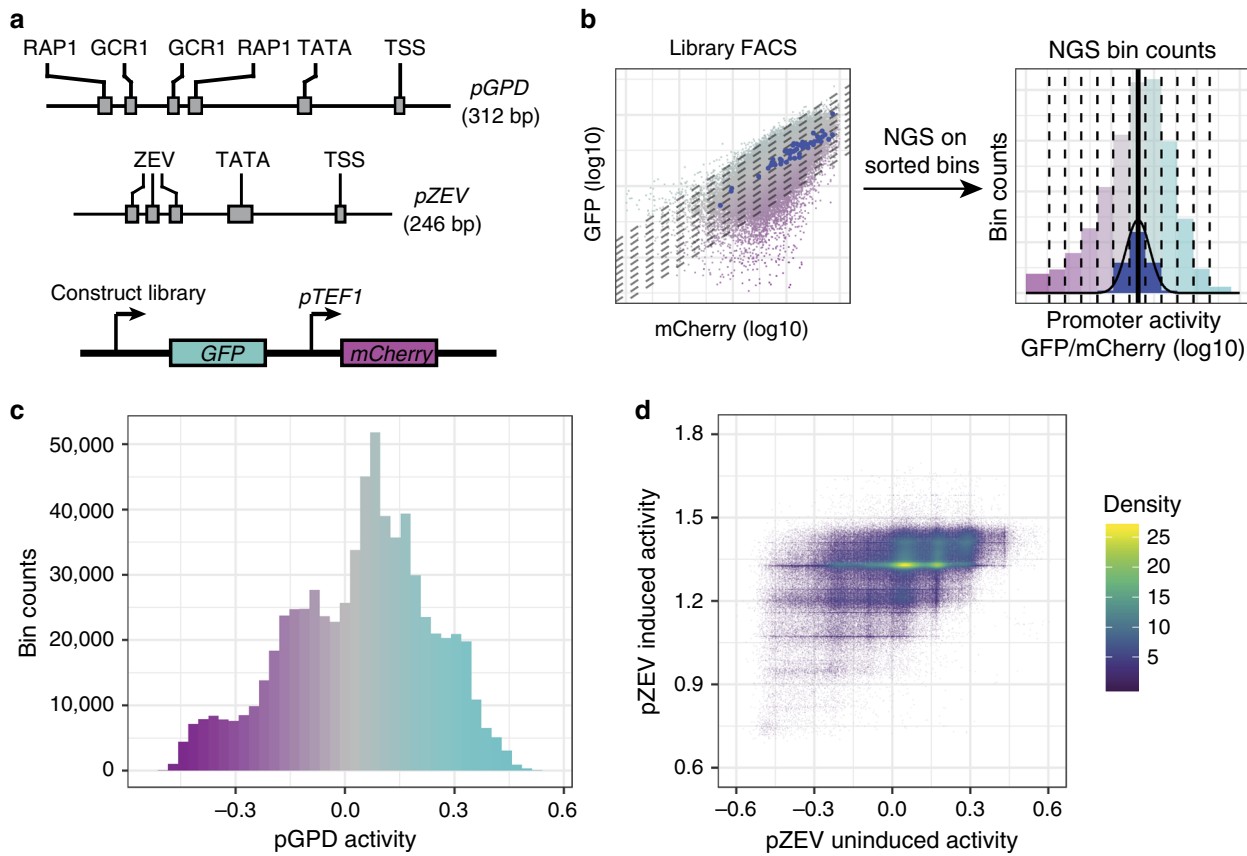

**Fig. 1 FACS-seq experimental strategy and data set overview.** **a** Schematic of tested libraries (above), indicating regions held constant in promoter design (gray boxes); schematic of two-color reporter device used to characterize promoter activity (below). RAP1, GCR1, ZEV transcription factor binding sites, TATA TATA box motif, TSS transcription start site motif. **b** Schematic of FACS-seq approach for high-throughput promoter activity characterization, in which next-generation sequencing (NGS)-derived histograms of sequence counts in FACS bins generated by sorting a library on promoter activity are used to derive promoter activity (log10 ratio of GFP to mCherry intensity, in arbitrary units) for each sequence in a library. Solid line: point estimate of promoter activity for an example sequence (blue points and histogram bins). Color gradient qualitatively indicates GFP:mCherry ratio for each cell or bin. **c** Histogram of promoter activities (log10 ratio of GFP to mCherry intensity, in arbitrary units) in the final $P_{GPD}$ library. Only sequences for which at least ten NextSeq reads were counted in each replicate were used in this analysis. Color gradient qualitatively indicates GFP:mCherry ratio for each sequence. **d** Density scatter plot of induced and uninduced promoter activities measured in the final $P_{ZEV}$ library. Only sequences for which at least 20 NextSeq reads were counted in each replicate were used in this analysis. Density: density of plotted points (arbitrary units).

18-bp TATA box. To maximize diversity, we proceeded with design 4 (3 ZEV sites, 37-bp internal spacer, 18-bp TATA box). Sequences in this library were 246 bp long, with 79% sequence randomized (Supplementary Fig. 6).

We measured individual sequence activities in the final libraries using FACS-seq[15]. Each library was sorted into 12 bins based on the GFP:mCherry ratio (Fig. 1b). Preliminary experiments and prior studies[15] indicated that this ratio is log-normally distributed. We define promoter activity as the base-10 logarithm of this ratio. Promoters from each bin were recovered, tagged with a bin identifier barcode, and analyzed through NGS. Count distributions across the bins were used to estimate individual promoter activities. Over 700,000 $P_{GPD}$ sequences (Fig. 1c) and 328,000 $P_{ZEV}$ sequences (Fig. 1d) were characterized. After outlier removal (see Methods), approximately 675,000 $P_{GPD}$ and 327,000 $P_{ZEV}$ sequences remained.

The $P_{GPD}$ FACS-seq was performed in duplicate. Data from the replicate experiments were consistent, with a coefficient of determination ($R^2$) of 0.94. Our final activity estimate was the mean of replicate activities. The $P_{GPD}$ data set contains sequences spanning promoter activities from −0.521 to 0.560 (median of 0.064, interquartile range of 0.273) (Fig. 1c, Supplementary Fig. 7). The $P_{ZEV}$ FACS-seq was performed in the presence and

absence of 1 μM beta-estradiol. Uninduced activities ranged from −0.76 to 0.62 (median of 0.04, interquartile range of 0.272). Induced activities ranged from 0.69 to 1.84 (median of 1.33, interquartile range of 0.122).

**Convolutional neural networks predict promoter activity.** We next built a model that predicts promoter activity using the data sets. We modeled activity as a function of sequence by implementing a CNN that accepts a DNA sequence as input and outputs an activity prediction, using the approach described in the Methods section.

The $P_{GPD}$ and $P_{ZEV}$ promoter designs are nearly identical (apart from the TFBSes and surrounding sequence), and deep learning performs best on very large data sets, including in genomics applications[37,38]. We hypothesized that the data sets could be merged and used to train a single model, taking advantage of the increased data set size. To further improve the model's performance, we trained an ensemble of submodels. All submodels used the same held-out test data (10% of the original data set), but the remaining data were divided equally into nine partitions, and each submodel used a different partition of these data for validation to prevent overfitting. For each submodel,

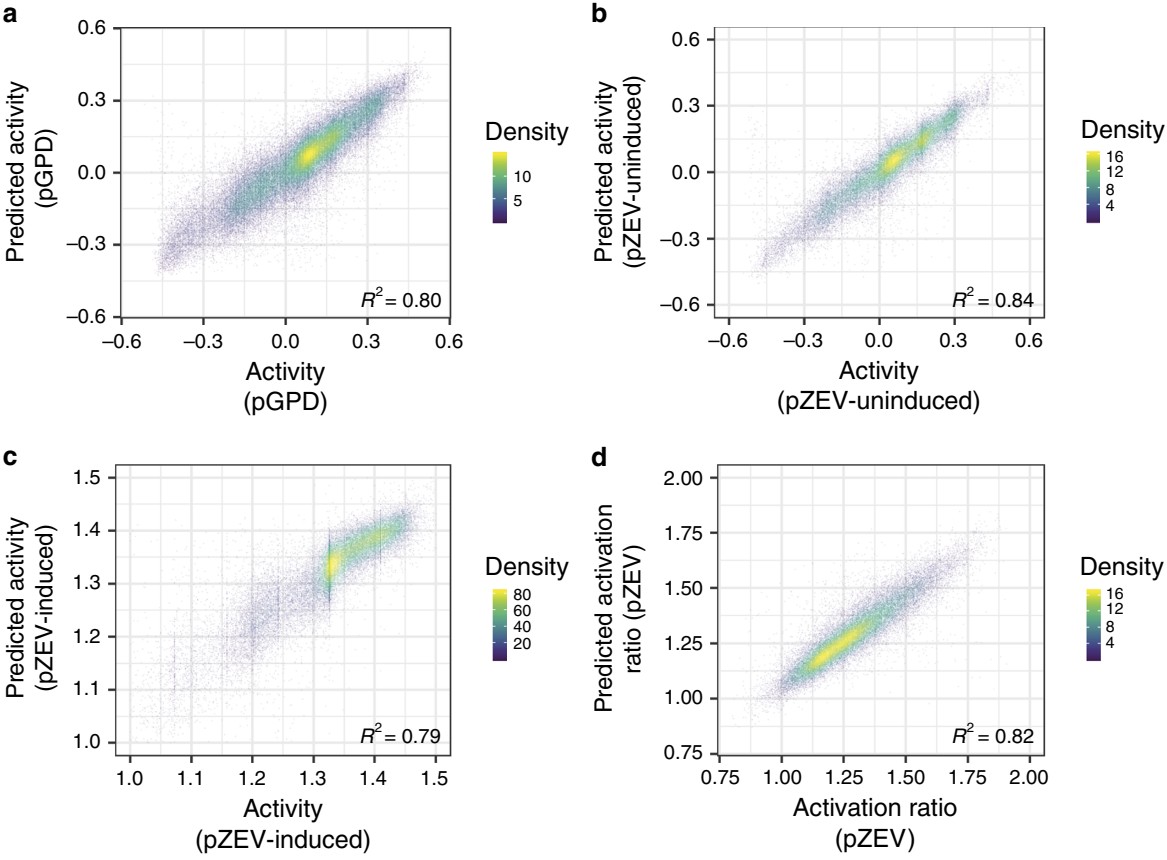

**Fig. 2 A neural network ensemble trained on P_GPD and P_ZEV data accurately predicts promoter activity.** Only sequences for which at least ten NextSeq reads were counted in each replicate were used in analyses of P_GPD data; only sequences for which at least 20 NextSeq reads were counted in each replicate were used in analyses of P_ZEV data. Density: density of plotted points (arbitrary units). **a** Predicted promoter activities versus FACS-seq measurements for P_GPD sequences in the held-out test data. **b** Predicted promoter activities in the uninduced condition versus FACS-seq measurements for P_ZEV sequences in the held-out test data. **c** Predicted promoter activities in the induced condition versus FACS-seq measurements for P_ZEV sequences in the held-out test data. **d** Predicted activation ratios (ratio of predicted induced and uninduced promoter activities) versus FACS-seq-derived activation ratios for P_ZEV sequences in the held-out test data.

training was interrupted after five epochs passed without improved performance on the validation data. Final predictions on test data were generated by averaging the submodels' predictions on the test data set.

We evaluated the model's performance by comparing model predictions to ground-truth measurements of P_GPD promoter activity (Fig. 2a), P_ZEV activity in the uninduced (Fig. 2b) and induced (Fig. 2c) conditions, and the ratio of induced to uninduced P_ZEV activity (Activation ratio, Fig. 2d). Model predictions generalized well to test data, achieving an $R^2$ of 0.80 for P_GPD activity, 0.84 for P_ZEV uninduced activity, 0.79 for P_ZEV-induced activity, and 0.82 for P_ZEV activation ratio. To determine whether merging the data sets and training a model ensemble improved predictions, we trained models with the same architecture but with only the P_GPD or only the P_ZEV data as input; prediction results for these models on the measures of promoter activity tested in Fig. 2 appear in Supplementary Fig. 8. Comparing the $R^2$ achieved by these models to the median $R^2$ achieved by the ensemble of submodels and to that of the final joined model showed that both merging the data sets and averaging the predictions of the model ensemble contributed to the performance of the final model, which outperformed the single-data set models on all prediction tasks shown in Fig. 2 (Supplementary Table 1).

**Model-guided design yields high-activity promoters.** To validate model predictions and generate promoters, we developed model-guided sequence design strategies. We designed promoters maximizing three objectives: P_GPD promoter activity, P_ZEV induced, and P_ZEV activation ratio.

Three sequence-design strategies were developed: screening, evolution, and gradient ascent. In the screening strategy, random sequences were generated following the specification of the original libraries, and accepted if their scores — predicted values for the objective property — exceeded a threshold (Supplementary Fig. 9). In the evolution strategy, mutagenized variants were generated from a candidate sequence. The variant with the highest predicted activity was accepted if its score exceeded the threshold. If not, a new set of variants was generated from this sequence, and the evolutionary cycle continued (Supplementary Fig. 10). The gradient ascent strategy (Supplementary Fig. 11) is a modification of the gradient descent process used to train neural networks[39,40]. It iteratively modifies initially random sequences by determining what incremental change would most increase the predicted score. These strategies are described in more detail in the Methods.

In addition to testing these design strategies, we tested applying an extrapolation penalty and/or a GC constraint when estimating promoter activities. Merging the predictions of the ensemble's submodels by taking their mean may allow outlier mispredictions to bias the final estimate. When using the extrapolation penalty, we merged ensemble predictions by computing the mean of predictions minus their standard deviation. Under the GC

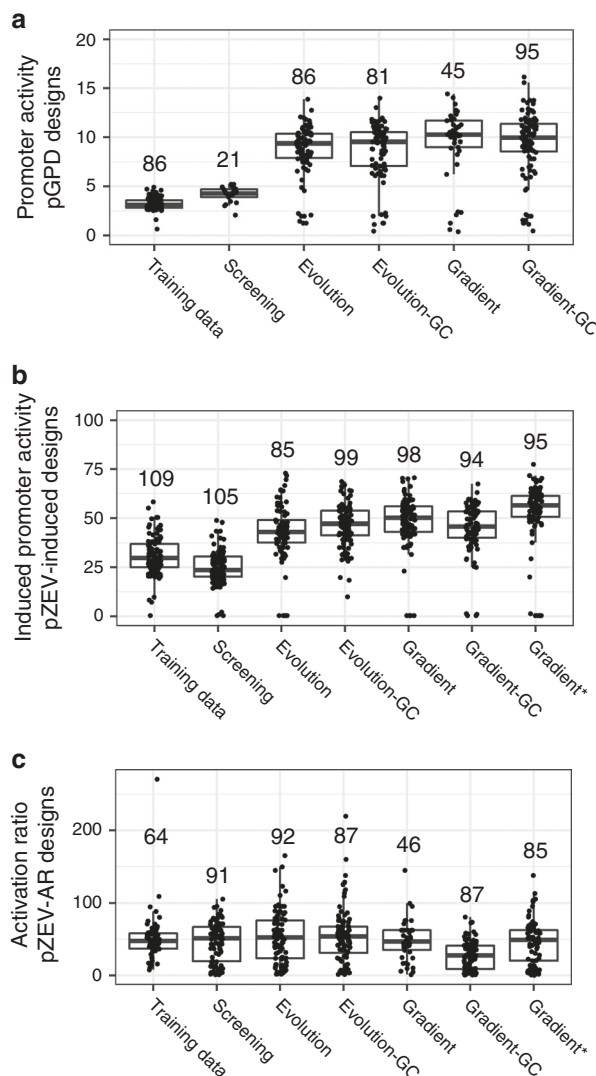

**Fig. 3 Performance of designed promoter sets in validation FACS-seq experiment.** In panels (**a**−**c**), boxes represent interquartile ranges; the bar within each box indicates the median. Whiskers extend to the furthest observation within 1.5 interquartile ranges of the nearest box edge. Numbers over boxplots indicate the number of sequences measured in FACS-seq in each promoter set. Promoter activities are shown here on a linear scale, and were transformed to a scale comeasureable with the results of individual promoter testing using a linear model fit to promoter activities measured by FACS-seq and by individual testing for a set of promoters spanning a range of expression activities. **a** FACS-seq measurements of promoter activities for $P_{GPD}$ promoter sets (or corresponding training data sequences). Training data: selected highly active sequences from the initial $P_{GPD}$ FACS-seq; Screening: $P_{GPD}$ promoter set generated using the screening approach; Evolution: $P_{GPD}$ promoter set generated using the evolution approach; Evolution-GC: $P_{GPD}$ promoter set generated using the evolution approach, with the GC constraint applied; Gradient: $P_{GPD}$ promoter set generated using the gradient ascent approach; Gradient-GC: $P_{GPD}$ promoter set generated using the gradient ascent approach, with the GC constraint applied. Points placed along the horizontal line were only measured in the highest-activity bin in FACS-seq. **b** FACS-seq measurements of promoter activities for $P_{ZEV}$ promoter sets designed to maximize induced activity (or corresponding training data sequences). Axis labels referring to $P_{ZEV}$-Induced sequences and designs, but otherwise as in (**a**); Gradient*: $P_{ZEV}$-Induced promoter set generated using the gradient approach, with an elevated target threshold set relative to other designs. Points placed along the horizontal line were only measured in the highest-activity bin in FACS-seq. **c** FACS-seq measurements of promoter activities for $P_{ZEV}$ promoter sets designed to maximize activation ratio (or corresponding training data sequences). Axis labels referring to $P_{ZEV}$-Activation Ratio sequences and designs, but otherwise as in (**b**). Source data are available in the Source Data file.

constraint, no 20-bp window in a sequence can have a GC content under 25% or over 80%. As extremes in GC content can complicate molecular biology procedures like PCR and Sanger sequencing[41], the GC constraint ensures promoter designs are tractable for downstream applications.

We designed 33 design approaches applying combinations of these techniques, as described in the Methods (Supplementary Table 2). Over 100 sequences were generated for each promoter set. Sequences from the original $P_{GPD}$ and $P_{ZEV}$ libraries were synthesized as controls to validate the data sets, examine whether prediction outliers resulted from errors in FACS-seq data, and compare the best-performing promoters in the original libraries to the model-designed promoters. The control promoter sets are summarized in Supplementary Table 3. We transformed the resulting library into CSY1252 and characterized the sequences using FACS-seq (Supplementary Fig. 12). We classified a sequence as successfully measured if the sequence was detected at least 20 times in the experiment. Two of the $P_{GPD}$ control sets and three of the $P_{GPD}$ design sets failed to meet this threshold (Supplementary Tables 2 and 3).

Control promoter set results were used to validate the original FACS-seq data for sequences with extreme activity values (outliers) and sequences whose activities were mispredicted by the model (inliers). Activity measurements in the original and the validation experiments for outlier sequences were poorly

correlated, in contrast to those for inlier sequences (Supplementary Fig. 13). Thus, outlier values from the original FACS-seq experiments were likely unreliable, supporting our decision to exclude these sequences from modeling. Sequences from nonoutlier control promoter sets were accurately measured (Supplementary Fig. 13, set 11, ZEV-grid), except that adjustments made to bin edges in order to measure highly active $P_{GPD}$ and $P_{ZEV}$-uninduced sequences reduced the sensitivity of mean measurements for low-activity $P_{ZEV}$-uninduced sequences. To determine whether our model-designed sequences outperformed the training data, high-activity control promoter sets were selected as training data benchmarks (Supplementary Table 3). Additionally, we fit linear models to the measured activities of $P_{ZEV}$ sequences measured in the original $P_{ZEV}$ FACS-seq and validation experiments in order to rescale the $P_{ZEV}$ means to correspond with validation results (Supplementary Fig. 14).

We next evaluated the performance of our design strategies. When measuring activities for $P_{GPD}$-Activity designs (Fig. 3a, Supplementary Fig. 15), in all cases, promoter sets using the extrapolation penalty exhibited significantly higher activities than corresponding designs where it was not applied ($p < 0.01$ for all, Mann−Whitney test (MWT)). We focused on the results for $P_{GPD}$-Activity promoter design strategies using the extrapolation penalty (Fig. 3a). Two sequences only appeared in the highest bin in one replicate; we excluded these from further analysis after verifying that this did not substantially affect our results. Screening promoters exhibited higher activities than training data sequences (median activity, screening: 4.29, training data: 3.13; $p = 1.18 \times 10^{-6}$, MWT). All measured Evolution and Gradient Ascent promoter sets exhibited substantially higher activities than Screening sets. The lowest median activity among the evolution and gradient ascent sets was 9.39 versus 4.29 for the screening designs. The gradient ascent strategy and the evolution

strategy generated designs with similar activities (without GC constraint, evolution: 9.39, gradient ascent: 10.27; with GC constraint, evolution: 9.55, gradient ascent: 9.97). The median activities were almost identical between corresponding promoter sets generated with and without the GC constraint (evolution: 9.55/9.39 with/without GC constraint; gradient ascent: 9.97/10.27 with/without GC constraint).

We next examined results for $P_{ZEV}$-Induced promoter designs (Fig. 3b, Supplementary Fig. 16). The extrapolation penalty significantly improved activities for almost all pairs of promoter sets tested with and without the penalty ($p < 0.01$ for all comparisons, MWT). We did not observe a statistically significant effect for promoter sets using the evolution strategy without the GC constraint (Supplementary Fig. 16).

We then focused on $P_{ZEV}$-Induced promoter sets using the extrapolation penalty (Fig. 3b). Eight sequences only appeared in the highest bin in the induced condition; we excluded these from further analysis after verifying that their exclusion did not substantially affect our results. Screening promoters generally exhibited lower activities than those in the training data (screening: 23.71, training data: 29.80, $p = 3.00 \times 10^{-6}$, MWT). However, all tested evolution and gradient ascent strategies generated promoter sets with substantially higher activities than training data. Applying the GC constraint did not result in a consistent impact on the model output; applying the GC constraint with the evolution strategy resulted in promoter sequences exhibiting higher activities (with/without GC constraint: 47.29/43.16, $p = 0.045$, MWT), whereas the opposite was observed with the gradient ascent strategy (with/without GC constraint: 45.90/50.18, $p = 0.012$, MWT). We also examined increasing the design threshold for a promoter set generated using the gradient ascent strategy with the extrapolation penalty and without the GC constraint. When increasing the target prediction value from 1.6 to 1.65, we observed an increase in median promoter activity (original threshold: 50.18, elevated threshold: 56.52, $p = 1.70 \times 10^{-5}$, MWT).

In contrast to the $P_{GPD}$-Activity and $P_{ZEV}$-Induced results, none of the median activation ratios for $P_{ZEV}$-Activation Ratio designs outperformed the training data sequences (one-tailed MWT, using a significance threshold of 0.05) (Fig. 3c, Supplementary Fig. 17). We determined that many of the sequences designed for this objective had uninduced activities at the lower limit of detection (i.e., cells containing these sequences were collected in the lowest-activity bin) (Supplementary Fig. 18). Specifically, 21.9% of training data sequences were at the lower limit of detection, significantly fewer than the 46.7−71.7% of designed sequences which were at the lower limit (Fisher's exact test; $p < 10^{-2}$ for all). This result suggests that our design strategies selected sequences with low uninduced activities to maximize the activation ratio.

Finally, to examine the designed promoters' sequence diversity, global alignment distances were determined for each pair of sequences within each promoter set. These alignment distances were used as a proxy for sequence diversity (Supplementary Fig. 19). Alignment scores in Screening and Evolution promoter sets were comparable to scores in corresponding training data sets. However, alignment scores in gradient ascent promoter sets exceeded scores in corresponding training data sets (MWT, $p < 10^{-49}$ for all). Thus, the screening and evolution strategies produced promoter sets with comparable diversity to the strongest promoters in the training data, while some diversity was lost when using the gradient ascent strategy.

**Individual characterization of designed sequences**. We characterized a subset of designed promoters using flow cytometry, to determine the reliability of measurements and compare our designed promoters to commonly used sequences. We selected sequences to test FACS-seq reliability across a range of activities, characterize sequences observed only in Bins 1 or 12, and to characterize sequences from promoter sets that were not characterized in FACS-seq, as described in the Methods.

We characterized ten randomly selected sequences from one promoter set for each of our design objectives (Final Promoter Set, Supplementary Table 2). As benchmarks, we characterized a set of commonly used promoters ($P_{GPD}$, $P_{TEF1}$, $P_{ADH1}$, $P_{PGK1}$, $P_{TPI1}$, $P_{CYC1}$) and three previously described ZEV promoters (P3, P4, P8)[10] (Supplementary Fig. 20). Activities and sequences for final promoters and controls appear in Supplementary Tables 4 and 5. All final promoter sets used the GC constraint and the evolution design approach. Altogether, we selected a set of 147 promoter sequences, of which 140 were successfully synthesized and characterized by flow cytometry, using the characterization construct (Supplementary Data). ZEV promoters were characterized in the presence and absence of $1 \mu M$ beta-estradiol. We compared activities successfully measured in FACS-seq to flow cytometry measurements (Fig. 4a); the two experiments' results correlated well ($R^2 = 0.92$).

We next examined flow cytometry results for the final promoter sets. For the $P_{GPD}$-Activity objective, in addition to the ten sequences from the final promoter set, we tested five sequences from three design strategies we were unable to test via FACS-seq (sets 22, 24, and 28 in Supplementary Table 2) to benchmark to control promoters (Fig. 4b, Supplementary Fig. 21). $P_{GPD}$ sequences from our selected set had activities ranging from 6.29 (95% confidence interval (CI) [5.31, 7.45]) to 13.36 (CI [12.06, 14.79]). The activities of $P_{TEF1}$ and $P_{GPD}$ were 1.52 (CI [1.45, 1.59]) and 14.65 (CI [13.46, 15.94]), respectively. Of the $P_{GPD}$ design strategies not previously tested, two produced similar results to our final strategy (Supplementary Fig. 21). However, of the five sequences synthesized from a promoter set which used gradient ascent to reach an elevated design threshold (set 28, Supplementary Table 4), one exhibited an activity of 20.17 (CI [17.12, 23.78]) — higher than that of the $P_{GPD}$ benchmark ($p = 1.43 \times 10^{-3}$, $t$ test). Thus, the gradient ascent strategy can produce promoters with higher activities than current best-in-class promoters.

We then examined how our final $P_{ZEV}$-Induced designs performed relative to ZEV benchmarks (Fig. 4c). Designed sequences' activities ranged from 27.03 (CI [24.77, 29.51]) to 63.85 (CI [53.30, 76.50]). All tested sequences exhibited higher activities than P4 and P8 ($p < 0.05$ for all, $t$ test), and most exhibited activities comparable to P3 (56.47 (CI [35.45, 89.95]). We reliably generated ZEV promoters with induced activities comparable to the best reported sequences.

Finally, we characterized sequences from our final $P_{ZEV}$-Activation Ratio design strategy (Fig. 4d). Measured activation ratios ranged from 90.75 (CI [76.37, 106.37]) to 258.73 (CI [235.88, 281.71]), while activation ratios for benchmark sequences were 542.20 (CI [348.14, 815.16]) for P3, 105.94 (CI [54.57, 195.10]) for P4, and 12.62 (CI [9.97, 15.67]) for P8. Examining the model-designed sequences' uninduced activities (Fig. 4e), we observed that uninduced activities for all but one were significantly below that of P3 ($p < 0.05$ for all, $t$ test), generally less than that of P4, and within the range of fluorescence levels observed from the no-GFP control plasmid (pCS4306, Background in Fig. 4e). While our designs' activation ratios did not surpass that of the benchmark sequence, we generated a large set of highly inducible sequences (over 100-fold), with particularly low uninduced expression.

**In silico mutagenesis reveals sequence design rules**. Finally, we identified features in designed sequences that contributed to

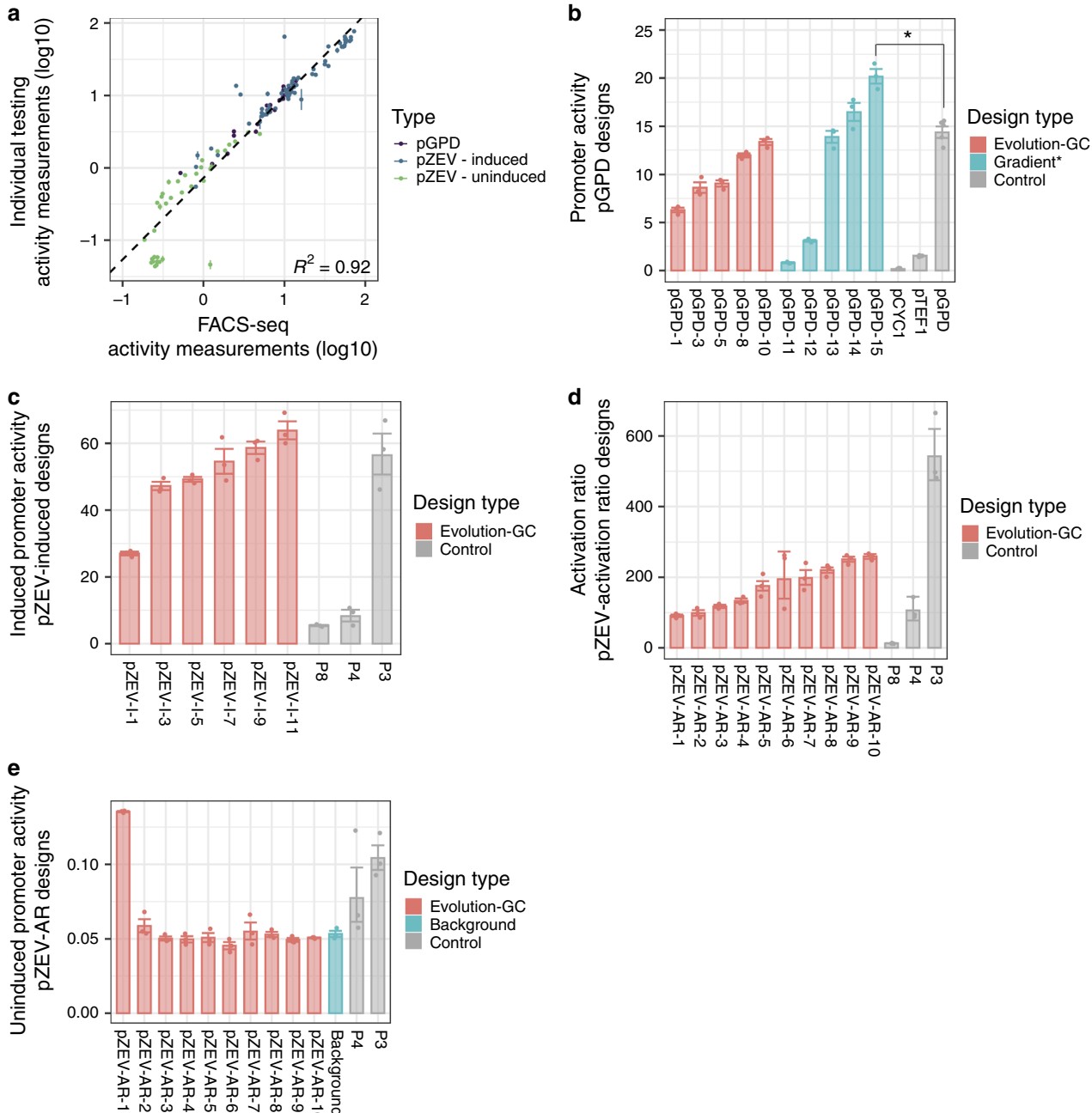

**Fig. 4 Validating activities of individual designed promoters by flow cytometry characterization. a** Promoter activity measurements (as base-10 logarithms) for selected sequences measured both in FACS-seq and by individual flow cytometry. FACS-seq: promoter activities determined from FACS-seq; Individual testing: promoter activities as determined by flow cytometry. Data are presented as mean values ± s.e.m. ($n = 3$ biologically independent samples). **b** Individually measured promoter activities (linear scale) determined by flow cytometry for selected $P_{GPD}$ designs and for control sequences (Control). Evolution-GC: randomly chosen sequences from the selected $P_{GPD}$ promoter set designed using the evolution strategy and the GC constraint. For clarity, a subset of those measured covering the measured range of activities is shown here; results for the entire set appear in Supplementary Table 4. Gradient*: randomly chosen sequences from the promoter set designed using the gradient strategy, with an elevated selection threshold. Control sequence names are indicated by text labels. *$p = 1.43 \times 10^{-3}$, two-sided $t$ test. **c** Individually measured promoter activities (linear scale) determined by flow cytometry for a selected $P_{ZEV}$-Induced design and for control sequences (Control). Evolution-GC: randomly chosen sequences from the selected $P_{ZEV}$-Induced promoter set designed using the evolution strategy and the GC constraint. For clarity, a subset of those measured covering the measured range of activities is shown here; results for the entire set appear in Supplementary Table 4. **d** Individually measured activation ratios (linear scale) determined by flow cytometry for a selected $P_{ZEV}$-Activation Ratio design and for control sequences (Control). Evolution-GC: randomly chosen sequences from the selected $P_{ZEV}$-Activation Ratio promoter set designed using the evolution strategy and the GC constraint. **e** Individually measured promoter activities (linear scale) in the uninduced condition for sequences displayed in (**d**) and for a Background control (pCS4306) expressing mCherry, but not GFP. In panels (**b**–**e**), promoter names, from left to right, are as in Supplementary Tables 4 and 5; bars and error bars in these panels represent the mean and s.e.m. ($n = 3$ biologically independent samples) of the original log-scale measurements, converted to linear scale. Source data are available in the Source Data file.

promoter activity. We examined one promoter set (In Silico, Supplementary Table 2) for each objective. For each set, we predicted all possible single mutants' activities. The mutagenesis score of each position in a sequence was defined as the greatest difference between the original sequence's score and that of the three corresponding mutants. We estimated each sequence position's relative importance by averaging position-wise scores across all sequences and normalizing them to the largest value, yielding a final position importance score (Fig. 5a).

For $P_{GPD}$ designs, important positions appeared throughout the sequence. For $P_{ZEV}$-Induced designs, the most important positions were in the spacer sequence at the 5′ end of the construct. For $P_{ZEV}$-Activation Ratio designs, these were immediately following the ZEV ATF binding sites. We examined the entire $P_{GPD}$ sequence set in more detail, but focused our investigation of $P_{ZEV}$ sequences on the identified regions.

We first generated sequence logos[42] for the $P_{GPD}$ sequence set (Fig. 5b). We observed a preference for certain bases immediately 5′ to the GCR1 motifs and following the TATA box. We also observed that T nucleotides were disfavored at the position 3 bases before the start codon, consistent with prior results[17]. We also observed this in sequence logos generated from both $P_{ZEV}$ designs (Supplementary Fig. 22a, b).

We then used in silico mutagenesis to uncover putative functional motifs in $P_{GPD}$ sequences. We scanned each sequence for contiguous blocks of six or more bases with mutagenesis scores under a target value chosen to select 5% of all randomizable bases, yielding a set of 177 candidate motifs. To explore these sequences' putative functions, we searched for TFBSes in these sequences using the Yeastract database[43]. Ninety-four sequences matched Yeastract TFBSes. All but 8 were 11 bp long or shorter; examining these, we found that at all lengths, predicted TFBSes were enriched relative to random sequences (Fisher's exact test, $p = 0.03$ or less for all). Of 121 transcription factors tested, matches to 16 were found. To account for overlaps in binding specificity, we pooled factors that always co-occurred together, yielding seven groups (Fig. 5c). This result suggests that these factors may be involved in transcription from the designed promoters.

In $P_{ZEV}$-Induced designs, position importance scores (Fig. 5a) and sequence logos (Supplemental Fig. 22c) indicated a role for AT-rich tracts near the 5′ end of each sequence. All sequences in the examined set contained a TA dinucleotide repeat at least 6 bp long in the 5′ spacer. As this motif resembles the TATA initiation motif, we speculated that the model may place this motif as an additional RNA polymerase site, although the TATA motif is located 3′ to TFBSes in natural promoters. Examining the longest TA repeat in each sequence, we calculated the median decrease in predicted promoter activity for a mutation within the repeat (the median score differential). Comparing these values to the scores of all bases in the $P_{ZEV}$-Induced 5′ spacers outside a TA repeat (Fig. 5d, Non-TA), we found that mutations within TA repeats had a greater predicted effect than mutations elsewhere (mean score differential $-0.071$ inside repeats, $-0.012$ elsewhere, $p < 2.2 \times 10^{-16}$, MWT). In comparing median score differentials for TA repeats of different lengths, the effect of a single mutation was greatest for 6-bp sequences and decreased with length (Fig. 5d; $p = 2.04 \times 10^{-5}$ or less for adjacent comparisons, MWT), possibly because a longer TA repeat provides redundancy against mutations.

Examining position importance scores (Fig. 5a) and sequence logos (Supplementary Fig. 22d) for $P_{ZEV}$-Activation Ratio designs, we observed that the ZEV ATF's binding motif, GCGTGGGCG, was frequently extended by the sequence GCTA. For each sequence, we determined whether the tetramer following each ZEV ATF binding site was GCTA, as well as the median score differential for these

sequences (Fig. 5e). Mutations in GCTA sequences had a larger impact on activation ratio than mutations in non-GCTA sequences (median score differentials in GCTA sequences $-0.080$, $-0.071$, $-0.132$ at ZEV ATF sites 1, 2, 3 vs. $-0.005$, $-0.007$, $-0.019$, in non-GCTA sequences; $p = 5.58 \times 10^{-15}$ or less for all, MWT). This suggests that appending GCTA to the ZEV ATF binding motif increases activation ratio, possibly by improving the site's binding affinity for the ZEV ATF.

## Discussion

We developed a CNN model that accurately predicts promoter activity in two yeast promoter libraries, and developed design strategies exploiting it to generate large, sequence-diverse promoter sets. To assay full-length promoters, we developed an FACS-seq pipeline that integrates data from two NGS platforms. We used a low-read, full-length sequencing run to determine each variant's sequence, and a high-read run covering only part of each variant to determine its abundance in each bin. In one recent work using an MPRA to characterize an 80-bp yeast promoter library, over 100 million sequences were measured at a low sequencing depth (78% of measured sequences had only one read assigned)[20]. By contrast, we chose to more precisely characterize a much smaller library of longer sequences, incorporating both constitutively active and inducible designs. Our model's ability to predict promoter activity for a complex sequence exemplifies the deep neural networks' ability to model complex data, building on previous work modeling sequence libraries in the 50-bp length range[23].

We found that designed sequences' properties varied depending on the design approach used. Screening sequences did not outperform the best sequences in the original data sets. However, when optimizing for $P_{GPD}$ activity and $P_{ZEV}$-Induced activity, the evolution and gradient ascent strategies generated sequences with activities comparable to or greater than benchmark promoters; we generated high-performing sequences even when applying a GC content constraint. These results demonstrate model-guided design's value in producing sequences with useful, rare properties. Generating new functional elements also tests a model's ability to generate accurate, generalizable predictions[24].

$P_{ZEV}$-Activation Ratio designs had lower apparent activation ratios than the benchmark P3 control promoter due to challenges in measuring very low activities accurately. Measured activities in the uninduced condition for $P_{ZEV}$-Activation Ratio designs were at the lower limit of detection, while P3 had a measurable level of uninduced expression (Fig. 4e). While activation ratio is an intuitive and simple measure of inducibility, it has limitations when applied to sequences with low uninduced activities. Developing a design strategy that independently optimizes for high induced and low uninduced activity may address this challenge. However, our designs did demonstrate a very low level of leaky expression, which may make them useful for applications requiring tight repression in the off state.

We also used the model to explore strategies that yielded promoters with high predicted activity or activation ratio. A similar in silico mutagenesis approach to ours was used previously to identify putatively functionally relevant features in natural promoters[44,45]. Because our model was trained on a data set of quantitatively characterized sequences designed using predetermined scaffolds, we were able to instead apply this approach to uncover sequence design strategies favored by the model and determine their predicted contributions to activity. The model's ability to identify motifs with a predicted functional role suggests that our approach may be valuable for future studies of native yeast promoter regulation; future studies could also investigate modifying the functional motifs that we held fixed, or

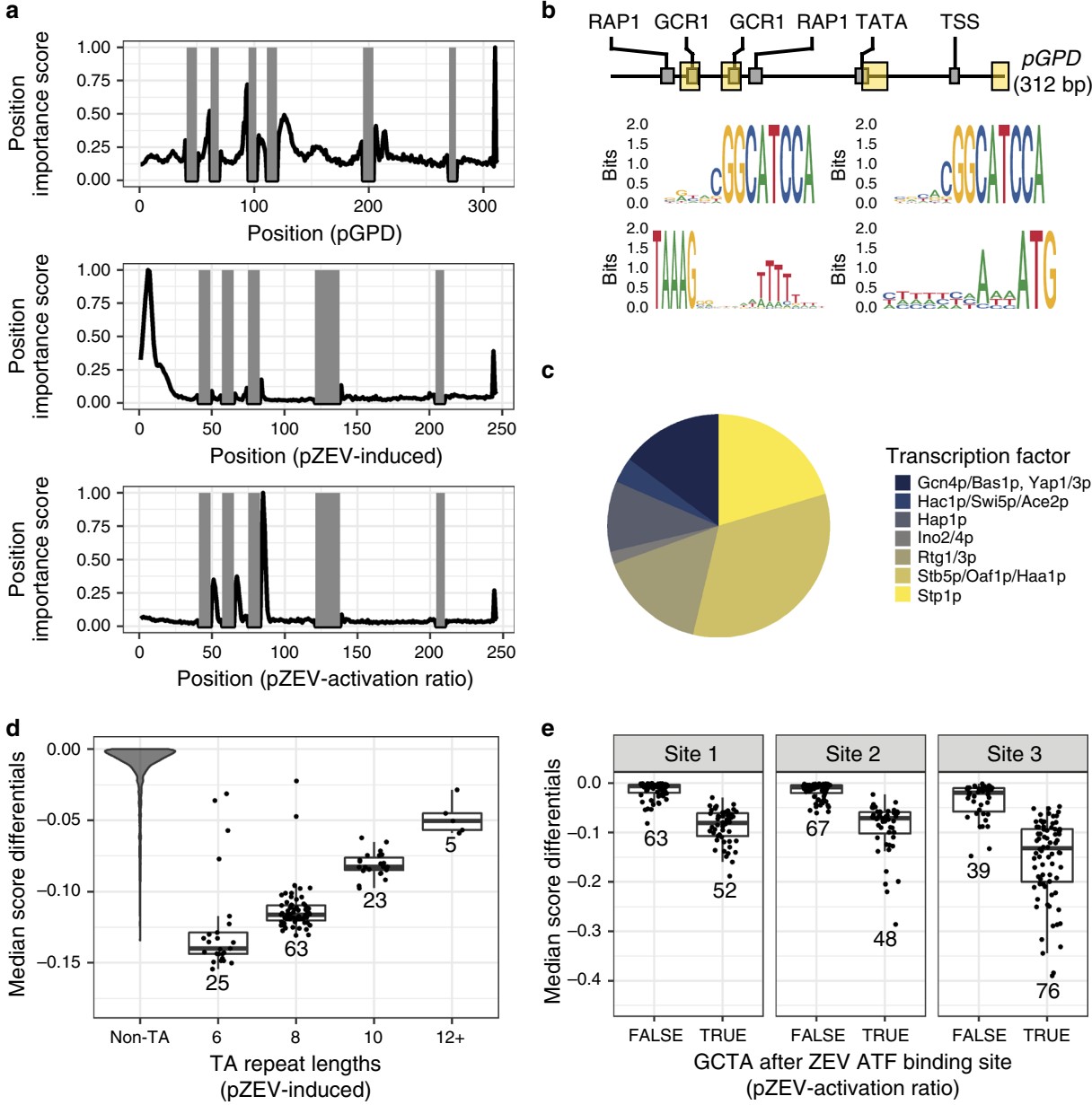

**Fig. 5 In silico mutagenesis enables identification of functional motifs in designed sequences.** In panels (**d**, **e**), boxes represent interquartile ranges; the bar within each box indicates the median. Whiskers extend to the furthest observation within 1.5 interquartile ranges of the nearest box edge. **a** In silico mutagenesis reveals patterns of predicted position-wise importance in designed sequences. Above: Normalized position-wise scores (averages of scores for each sequence in the design set) for a $P_{GPD}$ design set. Middle: Normalized position-wise scores for a $P_{ZEV}$-Induced design set. Below: Normalized position-wise scores for a $P_{ZEV}$-Activation Ratio design set. Gray areas: conserved regions, held constant. **b** Position-dependent features identified as sequence logos in $P_{GPD}$ designs. Above: Schematic of the $P_{GPD}$ construct (as in Fig. 1a). Highlighted regions: areas shown in detail below. Sequence logos, top left: Sequence logo for sequence context 5′ to first GCR1 site. Top right: Sequence logo for sequence context 5′ to second GCR1 site. Bottom left: Sequence logo for sequence context 3′ to TATA motif. Bottom right: Sequence logo for sequence context 5′ to ATG start codon. **c** Putative transcription factor binding sites identified in $P_{GPD}$ designs. Transcription factors with overlapping binding specificities were pooled as described in the text. Transcription Factor: factor or factors with a Yeastract motif matching the identified sequence. **d** Median score differentials of TA repeats in the 5′ spacers of $P_{ZEV}$-Induced sequences, by TA repeat length. Non-TA: score differentials for bases outside any TA repeat in the 5′ spacers of tested sequences. 12+: three sequences of length 12, one of length 14, and one of length 16. Numbers under boxplots indicate the number of sequences in each category. **e** Median score differentials of 4-bp sequences following each of the three ZEV ATF sites (Site 1, Site 2, Site 3) in $P_{ZEV}$-Activation Ratio sequences. TRUE/FALSE: Sequence after ZEV ATF site is/is not GCTA. Numbers under boxplots indicate the number of sequences in each category. Source data are available in the Source Data file.

test our hypotheses regarding the functional roles of the sequence features we identified.

Unexpectedly, the model placed TATA-like motifs at the 5′ end of $P_{ZEV}$-Induced designed promoters. While further characterization is needed, this approach may be of interest to future rational promoter design efforts. We additionally found that $P_{ZEV}$-Activation Ratio designed promoters often contained an apparent four-base extension (GCTA) of the ZEV ATF binding site. Although further characterization is needed, this extension may increase the sequence's binding affinity for the ZEV ATF, thus increasing ZEV

ATF-dependent transcription. Supporting this, previous studies have demonstrated a role in determining binding affinity for bases outside transcription factors' core motifs[46], including for the Zif-268 transcription factor used in ZEV ATF[47]. Thus, our approach could be applied to better characterize transcription factor-DNA sequence binding affinities.

By generating large promoter sets with activities comparable to or outperforming state-of-the-art benchmark promoters, our work creates a useful tool for applications in synthetic biology. Our approaches are not necessarily limited to yeast or to promoters, and could be applied to expression parts for use in other systems, such as mammalian cells, or to designing other classes of DNA sequences, such as terminators or RNA switches. Our results demonstrate that high-throughput characterization of artificial DNA sequence libraries enables accurate modeling of the DNA sequence−function relationship, which in turn enables the design of DNA sequences fulfilling challenging design constraints.

## Methods

**General**. Plasmids used in this study are described in Supplementary Table 6, and yeast strains used in this study are described in Supplementary Table 7. Expand High Fidelity PCR system (Roche Diagnostics) was used for PCR amplifications according to the manufacturer's instructions, unless described otherwise. Sanger sequencing was performed by Elim Biopharmaceuticals, Inc. All oligonucleotides used in this work appear in Supplementary Table 8. Plasmids and strains are available from the corresponding author upon request.

**Random library design and assembly**. Libraries were assembled by PCR-amplifying oligonucleotide libraries (Stanford School of Medicine Protein & Nucleic Acid Facility (PAN); Integrated DNA Technologies) corresponding to each spacer sequence, joining them via Golden Gate assembly[48] using constant sites located within constant regions, and PCR-amplifying the resulting products with primers providing 40 bp of homology to vector plasmids on each side. Oligonucleotides and PCR primers used to assemble libraries are listed in Supplementary Table 8; diversity was introduced by randomizing regions of the sequence outside the designed constant regions. The combinations of oligonucleotides used to PCR-amplify each fragment are given in Supplementary Table 9. PCR and Golden Gate reaction conditions are given in Supplementary Note 1; briefly, 15 fmol of each DNA fragment was used in 10-μl, 50-cycle Golden Gate reactions.

**Yeast library construction**. *Saccharomyces cerevisiae* strain CSY3 (W303 MATα) was used in the $P_{GPD}$ library experiment. To create a strain expressing the ZEV artificial transcription factor for the $P_{ZEV}$ library experiment, the $P_{ACT1}$ promoter and the ZEV artificial transcription factor gene were PCR-amplified in a single fragment from DBY19053 (ref. [10]), and a fragment containing the $T_{CYC1}$ terminator and the entire plasmid backbone was PCR-amplified from pCS2657 (ref. [49]). These fragments were joined by Gibson assembly[50], yielding pCS4339; the $P_{ACT1}$–ZEV ATF–$T_{CYC1}$ expression cassette was PCR-amplified and integrated into the *LEU2* locus of CSY3 using the Cas9-assisted integration method[51] (using pCS4187 (ref. [52]) as the guide RNA plasmid), yielding strain CSY1252. CSY1252 was also used in the promoter design validation experiments.

The plasmid pCS1748 (ref. [35]) expresses GFP and mCherry from separate copies of the $P_{TEF1}$ promoter. The plasmid pCS4305 was generated by digesting pCS1748 with *ClaI* and *MfeI* to remove the $P_{TEF1}$ driving GFP expression; a 675-bp sequence encoding $P_{GPD}$ and part of the 3′ UTR of YGR193C (Supplementary Note 2), and a sequence replacing a portion of GFP deleted by restriction digestion, were inserted using Gibson assembly[50]. The $P_{GPD}$ sequence was PCR-amplified from pCS2656 (ref. [49]), with bases added at the 3′ end to match the 5′ UTR of the S288C reference wild-type $P_{GPD}$ sequence[53]. To generate a vector for library integration, this plasmid was further modified by removing the $P_{GPD}$ promoter by digestion with *ClaI* and *MfeI* and using Gibson assembly to insert a sequence beginning with the first 60 bp of the YGR193C 3′ UTR sequence present in the original $P_{GPD}$ promoter (to ensure that all promoters were tested in a consistent genetic context). A *ZraI* cut site was created by adding a C nucleotide at the end of this 60-bp sequence, and the excised yEGFP sequence, lacking the first two bases of the yEGFP start codon, followed, yielding pCS4306 (Supplementary Fig. 23). To clone a promoter library into yeast, pCS4306 was linearized with *ZraI* digestion, and yeast were cotransformed with linearized plasmid and the library insert. This cotransformation was carried out as previously described[15]. Briefly, 50 ml yeast culture ($OD_{600}$ 1.3–1.5) was incubated with Tris-DTT buffer (2.5 M DTT, 1 M Tris, pH 8.0) for 15−20 min at 30 °C, pelleted, washed, and resuspended in Buffer E (10 mM Tris, pH 7.5, 2 mM MgCl₂) to 200 μl. To 50 μl of the yeast cell suspension, 2 μg of linearized plasmid and 1 μg of library insert DNA was added and the DNA-cell suspension was electroporated (2 mm gap cuvette, 540 V, 25 μF, infinite resistance). Transformed cells were diluted in 1 ml volume in yeast peptone dextrose (YPD) media, incubated for 1 h, then further diluted in yeast nitrogen base medium (BD Diagnostics) lacking uracil and containing 2% dextrose (YNB-U).

All libraries were grown in YNB-U, and passaged at least three times before sorting, with at least 10 $OD_{600}$*ml units transferred in each passage. For experiments involving $P_{ZEV}$ promoters, separate cultures with and without 1 μM beta-estradiol added were started 18 h before the sort. Cultures were back-diluted to an OD of 0.05−0.1 5 h before the sort to maintain them in log phase.

**Library sorting**. Cultures were harvested at an $OD_{600}$ of 0.7−0.8, spun down, and resuspended in phosphate-buffered saline (PBS) with 10 μg/ml DAPI (Thermo-Fisher). The sorts were performed on a FACSAria II cell sorter (BD Biosciences). Data was acquired using FACSDiva software (version 8.0.1), and excitation and emission filters for GFP, mCherry, and DAPI were as previously described[15]. Briefly, GFP was excited at 488 nm and measured with a splitter of 505 nm and bandpass filter of 525/50 nm, mCherry was excited at 532 nm and measured with a splitter of 600 nm and bandpass filter of 610/20 nm, and DAPI was excited at 355 nm and measured with a bandpass filter of 450/50 nm. Viable cells (as identified by a viability gate based on DAPI fluorescence and side-scatter area) were sorted into one of 12 bins of equal width on the basis of the GFP/mCherry ratio. These bins were chosen to cover the range of promoter activities present in the sorted library. The sort gates were generated using a MATLAB script. Supplementary Fig. 24 exemplifies this gating strategy.

Four bins were collected at a time, in three passes: one collecting bins 1, 4, 7, and 10, one collecting bins 2, 5, 8, and 11, and one collecting bins 3, 6, 9, and 12. For each pass, cells were collected until a target number of viable cells had been sorted. In the $P_{GPD}$ experiment, two replicates were collected. In experiments involving $P_{ZEV}$ inducible promoters, the 12 bins were each collected once for the uninduced and once for the induced condition. In these experiments, the gating and cytometry parameters were set separately for the uninduced and induced conditions. Counts of cells sorted per bin for the $P_{GPD}$ experiment are in Supplementary Table 10, for $P_{ZEV}$ in Supplementary Table 11, and for the validation FACS-seq experiment testing designed promoters in Supplementary Table 12.

The sort parameters used for the $P_{GPD}$ experiment were treated as reference conditions for experiments involving inducible promoters. To relate measurements from experiments involving inducible promoters to the results of the $P_{GPD}$ experiment, flow cytometry data collected for the libraries under the conditions used for sorting and under the reference conditions (those used to sort the $P_{GPD}$ library) were used as a benchmark to convert the GFP/mCherry ratios used as bin edges to their equivalents under the parameters used for the $P_{GPD}$ sort (Supplementary Fig. 25). Promoter activities were calculated for each cell, and approximately corresponding cells in each sample were identified by sorting these values. A linear model was fit, and bin edges used in the experiment as measured were converted to their equivalents under the reference conditions. For the validation FACS-seq, the fit was carried out using the mean of three samples collected under the experimental conditions and compared to one sample collected under the reference conditions.

**NGS sample preparation**. After sorting, cells were grown to saturation in YNB-U. 1.5 ml aliquots of cell culture were used as input in minipreps with the Zymoprep Yeast Plasmid Miniprep II kit (Zymo Research) according to the manufacturer's instructions. Multiple minipreps were performed where necessary so that one miniprep was performed for every 1,000,000 cells collected. In experiments involving inducible promoters, unsorted cells from the uninduced and induced libraries were also regrown and miniprepped. For each bin, the entire miniprepped volume was used as template in a PCR reaction using the KAPA HiFi PCR Kit (Kapa Biosystems). A tenfold dilution of this PCR product was used as template in a barcoding PCR adding Illumina adapter sequences and dual barcodes. As a quality-control measure, variable-length sequences were included in each primer immediately 5′ to the sequence annealing region, serving as a backup barcoding method. These PCRs were purified using the DNA Clean & Concentrator kit (Zymo Research) according to the manufacturer' instructions, quantitated using a Qubit fluorometer (ThermoFisher), and mixed at a ratio calculated to provide an approximately equal number of sequencing reads for each cell originally collected. This mixdown was then gel-extracted on a gel containing SybrSafe Red (ThermoFisher) and 2% agarose, PCR-amplified for 5−6 cycles starting from a concentration of 1 nM with primers corresponding to Illumina adapters to ensure full-length products, and purified, yielding the final NGS samples. PCR reaction parameters are given in Supplementary Note 1, oligonucleotides used in NGS sample prep are given in Supplementary Table 8, and oligonucleotide choices for the $P_{GPD}$, $P_{ZEV}$, and design validation FACS-seq appear in Supplementary Tables 13, 14, and 15, respectively.

Sample quality was checked using a Bioanalyzer 2100 (Agilent). Next-generation sequencing was carried out on an Illumina MiSeq by either PAN or the Chan Zuckerberg Biohub, using 2 × 300 paired-end reads, with PhiX sequencing control added to 30% by molarity to increase diversity in constant or AT-rich regions. In some experiments, the sample was additionally sequenced on an Illumina NextSeq by the Biohub, using 1 × 75 unpaired reads. Illumina control software version 3.0 was used for MiSeq experiments; version 2.1.0 was used for Nextseq experiments. FASTQ files were generated using Illumina bcl2fastq2 software version 2.20 with default parameters.

**Error-tolerant NGS sequence determination**. NGS data processing and model training were carried out on Google Cloud virtual machines, using Nvidia Tesla K80 GPUs.

Sequences in the sorted libraries were determined using the MiSeq output. Paired-end reads were first merged using Paired-End reAd mergeR (PEAR) version 0.9.6 (ref. [54]). In the $P_{GPD}$ and $P_{ZEV}$ experiments, there was a high risk that errors in PCR or mutations during passage after FACS sorting could give rise to similar sequences, differing at only a few positions. These sequences would likely have similar activities, and could lead to inflated estimates of model quality if one appeared in training data and another in validation or test data. To avoid this risk, we chose to group reads with similar sequences together and determine a consensus sequence. We sorted the sequences and compared each sequence with the one following using Needleman−Wunsch alignment, with a gap penalty of 5 and a mismatch penalty of 1. Needleman−Wunsch alignments were carried out using a parallelized implementation of the algorithm (https://github.com/hgbrian/nw_align/tree/54221ee). For each experiment, we established a cutoff value for similarity, and used this to determine where groups of related sequences began and ended (Supplementary Fig. 26).

Because sequences are arranged in alphabetical order in this process, mutations near the start of a sequence need to be accounted for separately. To do this, we repeated the process after reversing and re-sorting the original sequences; each read was thus assigned to two clusters, one from each sorting. Clusters with reads in common were then merged to yield the final groups of reads corresponding to each sequence. The distribution of the numbers of reads in each cluster in the final $P_{GPD}$ and $P_{ZEV}$ libraries appears as Supplementary Fig. 27.

Read clusters were reduced to consensus sequences by taking a majority vote at each position in the sequence to obtain a consensus call for that position. Singleton sequences and sequences without a majority call at each position were discarded (Supplementary Fig. 27). To verify that this procedure produced clearly defined read clusters, we sampled 400 clusters from each data set. For each cluster, we calculated the longest alignment distance between any pair of sequences within the cluster, as well as the shortest alignment distance to any other cluster in the data set. We found that the resulting distributions of alignment distances showed good separation for both data sets (Supplementary Fig. 28).

**Measuring promoter activity**. In the $P_{GPD}$ and $P_{ZEV}$ experiments, the MiSeq runs yielded 0.5 or fewer reads per original cell for most bins. For many sequences, there were enough reads to identify the sequence itself, but not enough to accurately quantitate promoter activity. To solve this problem, the samples were resequenced on an Illumina NextSeq using a $1 \times 75$ single read kit (for lack of paired-end kits long enough to sequence this sample in its entirety on NextSeq). This allowed many more sequences to be accurately quantitated — the distribution of read counts in the raw NGS data for the final $P_{GPD}$ and $P_{ZEV}$ libraries appears as Supplementary Fig. 29. Measures of promoter activity derived from NextSeq data and full promoter sequences derived from MiSeq data were related using the first 35 bp of each library member, which is fully randomized and was found to act as a unique identifier for almost every sequence. In the validation experiment, the MiSeq data was used to calculate promoter activity directly, since this experiment featured a relatively small number of designed sequences.

Promoter activities were obtained following previously described methods[15,16]. First, each read was assigned to a bin by identifying the barcoding oligos used to generate it; each oligo contained either a variable-length skew sequence read as part of the sequencing read or a barcode, determined in a separate barcoding read. Sequences with fewer than a threshold number of reads measured in each replicate were discarded. The choice of skew sequences and barcodes used to assign reads to bins in each experiment, and the read count thresholds used in each experiment, are provided in Supplementary Table 16. Read counts in each bin $i$ were then normalized by multiplying by $C_i/R_i$, where $C_i$ is the number of cells collected in bin $i$, and $R_i$ is the total number of reads observed in the bin. A maximum-likelihood estimation process was used to assign a mean to each sequence. As much of this process as possible was executed in parallel on the GPU, using the Numba project's CUDA libraries (numba.pydata.org).

When estimating means for each sequence, each replicate of each experiment was processed separately. Based on prior experience[15] and the results of preliminary experiments, it was assumed that the distribution of fluorescence for each sequence in a library was log-normal, and that the standard deviation of the fluorescence distribution $\sigma$ was the same for all sequences. To provide some robustness against outliers, it was further assumed that with a probability $\varepsilon$, cells were collected not from the log-normal distribution, but from a uniform distribution across all bins.

The mean estimation process then requires two hyperparameters: $\sigma$ and $\varepsilon$. For a given promoter activity $\mu$, a given $\sigma$ and $\varepsilon$, and a number of bins $N$, the probability of a cell being observed in a bin $i$ with edges $a_i$ and $b_i$ is

$$P_{\mu,\sigma,\varepsilon}(i) = \left[F_{\mu,\sigma}(b_i) - F_{\mu,\sigma}(a_i)\right](1-\varepsilon) + \frac{\varepsilon}{N}, \tag{1}$$

where $F_{\mu,\sigma}$ is the cumulative distribution function of a normal distribution with mean $\mu$ and standard deviation $\sigma$. Supposing that the number of cells observed in bin $i$ is $r_i$, the log-likelihood of a set of parameters given observed data is[16]

$$\sum_{i=1}^{N} r_i \times \log(P_{\mu,\sigma,\varepsilon}(i)). \tag{2}$$

Let $(\sigma^*, \varepsilon^*)$ be a choice of values for $(\sigma, \varepsilon)$. For a set of $M$ possible values of $\mu$, we can construct an $N \times M$ matrix

$$W_{(\sigma^*,\varepsilon^*)}, \text{ where } W_{(\sigma^*,\varepsilon^*)ij} = \log\left(P_{\mu_j,\sigma^*,\varepsilon^*}(i)\right). \tag{3}$$

Supposing there are $S$ sequences total, if the $S \times N$ matrix $A$ contains the number of cells observed in each bin for each sequence, the product $AW$ is an $S \times M$ matrix giving the log-likelihood of each possible value of $\mu$ for each sequence. Estimates of $\mu$ are chosen for each sequence to maximize log-likelihood, and the sum of the log-likelihoods of each sequence acts as a score for the original choice of $\sigma$ and $\varepsilon$.

Using parallel computation on a GPU to accelerate this process enabled us to optimize the hyperparameters used in fitting via a grid search, to determine the sensitivity of fits to hyperparameter choice. Optimal hyperparameter values were found for each replicate or condition in each experiment (Supplementary Fig. 30, Supplementary Table 17). Additionally, we tested the sensitivity of the fit to the choice of hyperparameter scores, by calculating the root-mean-squared distance from the vector of fit means under the final hyperparameter values to the vector of fit means under each other choice of hyperparameters tested (Supplementary Fig. 30).

This resulted in a table of sequences and promoter activity values — for the $P_{GPD}$ experiment, promoter activity was measured once in each replicate, while in the $P_{ZEV}$ and the validation experiment, it was measured once in the uninduced and once in the induced condition. Sequences with mutations in the designed constant regions were removed. In the $P_{GPD}$ experiment, sequences with replicate promoter activity estimates differing by over 0.2 or which were only observed in the highest or lowest bins were considered outliers. In the $P_{ZEV}$ experiment, sequences for which all observed reads fell in either the highest or lowest bins in either the uninduced or the induced condition were considered outliers. The resulting table was then used to train models of promoter activity.

**Model implementation and training**. Models were implemented in Keras (keras.io) version 2.1.6, with Tensorflow version 1.7.0 as the backend. Following the approach of Kelley et al.[22], sequences were encoded in one-hot format; each sequence was represented as a two-dimensional matrix with four rows, one for each base of DNA, and a column for each position. The values of the matrix at each column were 1 in the row corresponding to the base present at that position, and 0 elsewhere. The model consists of a series of convolutional and max-pooling layers, followed by a two-layer fully connected network, which outputs a single prediction of promoter activity. All convolutional layers had a width of 8 and 128 output channels and used a rectified linear unit (ReLU) nonlinearity, with batch normalization applied between the convolution and the ReLU layer. Input sequences were processed with six rounds of convolution and max-pooling with a stride length of 2. Two 128-unit fully connected layers, each followed by batch normalization and a ReLU, were then applied, followed by a final fully connected layer which directly provided the final output(s). In order to generate separate predictions for both the uninduced and the induced $P_{ZEV}$ conditions in one model, we used an output layer with two units, weighting the loss on both predictions equally. L2 regularization with a weight of $10^{-4}$ was applied at all layers except the final output.

A training/validation/test split of 80%:10%:10% was used in training the models. All sequences were padded on each side with at least 25 bp of the surrounding pCS4306 vector sequence; to account for the different lengths of $P_{GPD}$ and $P_{ZEV}$ promoters when training models on the merged data sets, the pads were extended for $P_{ZEV}$ promoters, to a total length of 58 bp on each side. During training and validation, the data set was augmented by applying a shift of $0-7$ bp.

Models were trained with Adam as the optimizer[55], with a learning rate of $10^{-5}$. Huber loss with $\delta = 0.15$ was used as the loss function. When the model had two outputs, both were weighted equally in calculating the loss. Model training was terminated using early stopping after five consecutive epochs with no improvement in validation loss.

**Promoter design strategies**. The strategies tested for designing promoters are summarized in Supplementary Table 2. Three objectives were optimized: overall activity for $P_{GPD}$ designs, activity under beta-estradiol induction for $P_{ZEV}$ designs, and activation ratio for $P_{ZEV}$ designs. In all cases, at least 100 promoters with the objective predicted to be above a design threshold were generated. The design thresholds were chosen empirically to generate the desired promoter sets without expending an unreasonable amount of computational resources.

The predictions for each sequence were generated using the set of nine models trained on the joined data sets. Predictions from the models were merged either by taking the mean of the individual predictions, or the mean of predictions minus their standard deviation (to buffer the possible effect of outlier predictions). To increase the diversity and ease of assembly of the designed sequences, a filter was applied in some experiments to reject sequences containing regions 20 bp or longer with GC content below 25% or above 80%.

As described in "Results", three design strategies were tested: screening, in silico evolution, and gradient ascent. In screening, sets of sequences were randomly

generated, using the same constant regions and base composition probabilities used in designing the original libraries. These sequences were then tested and accepted if they met the objective. In in silico evolution, a set of sequences was iteratively generated from a randomly chosen parent sequence and tested; the highest-scoring sequence was passed on to the next round of evolution. The number of mutations induced in each round decreased over time. We implemented a gradient ascent strategy for promoter design by calculating the gradient on the input data, with respect to the score property to be maximized. Given a one-hot encoded DNA sequence, the gradient was iteratively calculated and used to generate an updated version of the input. The objective score was calculated for a rounded one-hot matrix derived from this updated input. The sequence was accepted if its score was above a threshold. Values of cycle-dependent parameters used in the evolution and gradient-ascent strategies are given in Supplementary Tables 18 and 19, respectively.

The score threshold for accepting a design was set independently for each of our three objectives. For most evolution and gradient ascent designs, it was set such that a promoter set of 120 sequences could be generated within an hour for a selected reference design (Supplementary Table 2, Threshold Reference in Notes). The screening strategy was unable to generate promoter designs that reached this threshold; so for screening designs, this threshold was decreased in increments of 0.05 until a promoter set could be generated. Additionally, an elevated-threshold design was generated for each objective using the gradient ascent strategy and the extrapolation penalty, not applying the GC constraint, and increasing the threshold in increments of 0.05 until a set of 120 sequences could no longer be generated within an hour. Sequences containing *Bsa*I restriction sites, which interfere with the assembly strategy, were removed. The choices of parameters used to specify each experiment (target promoter, objective to maximize, use of optional GC filter, function used to merge submodel outputs, design strategy, final score threshold) are provided in Supplementary Table 2.

**Assembling designed sequences from an oligo pool.** Sequences derived from the sequence evolution strategies or selected from the original FACS-seq data as controls were assembled from an oligonucleotide pool (Twist Bioscience). Each sequence set was assigned unique PCR amplification sites, designed using a Python script to minimize cross-talk between pools; oligo annealing temperatures in this script were calculated using the Primer3 library[56]. (All Python scripts were run using Python 2.7 on Ubuntu 16.04.) Each sequence was then designed as a pair of oligos (a forward and reverse oligo), which could be joined by Golden Gate assembly using a unique assembly site.

Forward and reverse oligos for each sequence in each sequence set were amplified from the oligonucleotide pool in KAPA PCR reactions, using selective primers complementary to the designed unique PCR amplification sites to selectively amplify the desired subpool. The designs for each sequence set were then assembled by Golden Gate assembly (following the reaction conditions described in Supplementary Note 1 as "Golden Gate Assembly of Libraries") and further PCR-amplified. An equimolar mixture of the resulting subpools was then cloned into pCS4306 by gap repair as described above.

**Testing individual sequences.** To validate FACS-seq results and further characterize designed promoters, a subset of sequences was chosen to be synthesized and tested individually. Sequences were chosen using an R script to obtain a minimal set of sequences needed to test hypotheses of interest. To determine FACS-seq measurement reliability across a range of activities, we selected three sequences for each of eight evenly distributed FACS-seq-derived activity values, in both the uninduced and induced conditions. To better characterize sequences that were observed in an extreme bin (Bins 1, 12), for each promoter set, we selected up to three sequences observed only in an extreme bin in either the uninduced or induced condition. Finally, we randomly selected sequences from each promoter set that was not characterized in FACS-seq, such that 5 sequences were measured in total.

To clone single promoters, the sequences were ordered from Twist Bioscience, or in the case of pre-existing control promoters, PCR-amplified using Expand High Fidelity PCR from plasmids: $P_{GPD}$ and $P_{TEF1}$ from pCS4305, $P_{ADH1}$ from pCS2660, $P_{PGK1}$ from pCS2663, $P_{TPI1}$ from pCS2661, $P_{CYC1}$ from pCS2659, P3 from pCS4307, P4 from pCS4308, and P8 from pCS4309. Plasmids pCS4307, pCS4308, and pCS4309 were constructed by digesting pCS4306 with *Zra*I and using Gibson assembly to insert the corresponding ZEV promoter sequence, which was amplified from gDNA of a yeast strain containing the sequence (DBY19053 for P3, DBY19059 for P8) or (in the case of P4) artificially synthesized as a gBlock Gene Fragment (IDT). We were unable to PCR-amplify the P4 sequence from gDNA, or have it synthesized as originally specified; we replaced the second of six closely spaced GCGTGGGCG sites in the original sequence with TTACTCAAG. Sequences were cloned into pCS4306 by gap repair using the Frozen-EZ Yeast Transformation II Kit (Zymo Research) according to the manufacturer's instructions. Colonies were inoculated into 500 µl YNB-U liquid media in 96-well plates and grown with shaking at 30 °C overnight; 5 µl of the resulting seed cultures were used to inoculate new 500 µl sample cultures. Each sample culture was derived from a unique transformant colony. These were assayed on a MACSQuant VYB flow cytometer (Miltenyi Biotec GmbH) after 24 h further growth. We performed the manufacturer's recommended calibration procedure before each flow cytometry run. For each sample, the ratio of measured GFP to mCherry fluorescence was determined for all detected events, and the median of these values was used as the measure of promoter activity in the sample. Three biological

replicates were tested for each sequence. Confidence intervals were calculated using a *t* test with the appropriate degrees of freedom for each sample. Data were confirmed to be normally distributed (conditional on the sequences tested) using a Q−Q plot (Supplementary Fig. 31).

**Motif identification by in silico mutagenesis.** Single mutants of sequences to be characterized by mutagenesis were generated, and activities and activation ratios estimated, using a Python script. We used a Python script to calculate score differentials between activity predictions for double mutants and single mutants, as described above under the heading "In silico mutagenesis reveals sequence design rules", as well as to identify strong motifs in $P_{ZEV}$-Induced and $P_{ZEV}$-Activation Ratio designed promoters.

**Statistics.** Statistical significance was established using the Mann−Whitney test, Student's *t* test, or Fisher's exact test, as indicated in the text. All tests, as well as calculations of the coefficient of determination ($R^2$), were carried out using R scripts, using default settings. R scripts were run under R 3.5.1 and developed using RStudio 1.1.463 for Windows 10. All tests were two-tailed except where stated otherwise. Replicates were defined as cultures inoculated from separate yeast colonies or streaks and cultivated in separate containers.

**Reporting summary.** Further information on research design is available in the Nature Research Reporting Summary linked to this article.

## Data availability
NGS data that support the conclusions of this study have been deposited in the NCBI Gene Expression Omnibus (GEO) with the accession code GSE135464. All other data is available in Zenodo (https://doi.org/10.5281/zenodo.3735426), including R scripts and raw data for regenerating all data figures. The source data underlying Figs. 3, 4, 5c−e and Supplementary Figs. 15–18, 20, and 21 are provided as a Source Data file.

## Code availability
All code produced in this study is available via Github (https://github.com/smolkelab/promoter_design), under the MIT license.

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

## Acknowledgements

We thank Deze Kong, Peter Dykstra, Aaron Cravens, and Prashanth Srinivasan for valuable comments on the manuscript. We thank Aaron Cravens for assistance with FACS sorts and for providing Cas9 plasmid pCS4187, and Travis Horst for assisting in the construction of strain CSY1252 and plasmids pCS4307-9 and pCS4339. Yeast strains DBY19053, DBY19054, and DBY19059, containing components of the ZEV system, were generously provided by Dr. David Botstein and Dr. Patrick Gibney. This work was supported by the National Institutes of Health (grants to C.D.S.) and Chan−Zuckerberg Biohub (award to C.D.S.), as well as the Stanford Bio-X Institute and the Siebel Scholars Foundation (graduate student fellowships to B.J.K.).

## Author contributions

B.J.K. and C.D.S. designed the project and wrote the article. B.J.K. performed experiments and conducted data analysis.

## Competing interests

The authors declare no competing interests.
