## [Peer Review File · Nature Communications]

Reviewers' Comments:

Reviewer #1:

Remarks to the Author:

General Comments:

This study focused on promoter engineering in the yeast *Saccharomyces cerevisiae*. The strong-constitutive GDP and orthogonal inducible pZEV promoters were randomised in spacer regions between known transcription factor binding sites TATA boxes, and transcription start sites. These promoters were used to control GFP expression, which was normalised to constitutively expressed mCherry in each cell to control for noise. Yeast populations transformed with diversified promoter libraries were then FACS sorted using 12 gates that spanned the range of GFP to mCherry values. Plasmid DNA was extracted from sorted populations and next-gen sequenced. Deep learning was then used to model the relationship between promoter sequence and GFP:mCherry expression. Predicted expression levels correlated well with actual measurements of those sequences.

The model was then successfully used to predict novel sequences with higher expression levels than any sequence in the library or even the wild-type promoter sequences. Importantly, different methods of designing sequences were compared, with evolutionary optimisation found to give both high expression and high sequence diversity. This is important for future application of designed sequences as it minimised their homology and therefore loop-out potential when used multiple times in yeast. Another interesting finding was that the activation ratio of the inducible pZEV promoter could not be optimised because of an abundance of promoter sequences at the lower limit of detection in the training data. When individual promoters were tested alongside controls, the gradient ascent strategy was found to enable the design of a promoter with ~30% higher output than the native GDP promoter.

Finally, the importance of each base position was analysed alongside putative transcription factor binding sites and TATA boxes that may have been introduced in random sequences. Interestingly, new TF binding sites were significantly enriched in the high-expression sequences, as was a novel TATA box placement.

The study is interesting, thorough, novel, well-written, and of high fundamental importance. However, the results are underwhelming from an applied perspective given that only marginal improvements in expression of designed promoters could be achieved (or none except for leakiness in the case of pZEV) (Figure 4). My interpretation of this is that machine learning has limited potential to predict and engineer improved promoter designs, especially relative to other more common approaches. However, this could be due to the promoter library design, which left core promoter motifs unaltered.

I have no serious concern with the technical and scientific content of the article, but mainly question whether the results are impactful enough for Nature Communications. Regardless, the comments below are made out of genuine curiosity, and might be useful for improving the manuscript.

Please note that I am not qualified to review the deep learning methodology. However, given the predictive power of the model and the empirical follow-up experiments I am unconcerned with this.

Major comment

1. Why is there such a large discrepancy between the improvement in GDP promoter expression seen in the designed promoters in Figure 3-a and that seen from the results in Figure 4-b? In 3a there is a doubling in expression level relative to the high expression-level training data while in Figure 4-b there is a ~30% improvement over the control promoter.

In the methods section, the protocol used for measuring individual promoter activities (section 4.10) seems as though it may result in growth-stage variation between different strains. There is no control of inoculation density between the preculture and the main culture, which is not necessarily problematic, except that the main culture is grown for 24 hours prior to flow-cytometry measurement. Growth stage has a significant effect on promoter activity, and could explain the above discrepancy. Even if all cultures and replicates are in the same growth stage, it is uncertain whether they are in exponential growth, post diauxic shift, or stationary phase when the readings were taken. This could also explain why the TDH3 promoter has a 7-fold higher activity than TEF1 (Figure 4 b), which contradicts previous reports where pTEF1 activity is typically half of pTDH3.

Minor comments

2. How was diversity introduced to the promoter libraries during construction? The methods state that oligos corresponding to each variable region were PCR amplified and golden-gate assembled to make the promoter libraries. Was variation introduced solely through synthesis errors in the oligo pool? If so, then please make this clearer; if not, then please explain how genetic diversity was constructed. In general, some more detail on the library construction would be useful.

3. I would expect that there would be increased GFP:mCherry variation in the promoter libraries relative to native promoter sequences. Did the distribution of GFP:mCherry values change in the diversified promoter libraries relative to the wt promoter sequence?

4. During library construction, transformation, and outgrowth, were there any attempts to estimate the library size, or is the size estimate solely from sequencing?

5. Line 262 refers to a 'one-hot encoded dna sequence'. I thought this was a typo intended to be 'one-pot'. A brief explanation or contextualisation of 'one-hot' would make this section easier to digest for a broad readership.

6. What was the overall variant frequency for individual sequences in the promoter libraries?

7. How was auto-fluorescence controlled? How were experiments conducted on different days normalised and compared?

8. In the discussion it would be useful to contextualise and compare these results with other similar and recent efforts. For example, "Deciphering eukaryotic gene-regulatory logic with 100 million random promoters" published in Nature Biotechnology in December 2019.

Reviewer #2:

Remarks to the Author:

This work is an interesting extension of the previously published research modeling yeast MPRA datasets of artificial 5' UTRs using CNNs (Genome Res. 27, 2015–2024 (2017) and Nature Biotechnology volume 37, 803–809 (2019) to analysis and modeling the whole promoter sequences. The article can be considered as an important evaluation of model-guided promoter design strategies and promoters' characteristics that are responsible for their activity.

The authors characterized their models performance by R2 using 10% of the dataset for the test. However,

it might be that their training set had promoters highly similar by sequence with the test set examples and they predicted functional activity based mostly on sequence similarity (in such case the model will not work for promoters that provide high expression but very different by sequence from the training set). Ideally, they should get rid of highly similar sequences between their training and test set, then they can estimate the actual performance of their model in application to unseen sequences.

CNN models for promoter identification in genomic sequences have been presented in PLoS One. 2017; 12(2): e0171410 and its following paper in Bioinformatics, 35 (16), 2730-2737 2019 describing possibility to reveal promoter functional sub-regions by in-silico mutation experiments. It would be interesting to see a discussion how the experimental technique applied in the paper increases the discovery power of pure computational approaches.

The authors claim (to provide a tool for generating promoters with user-specified functional properties) is an overstatement as in the article they just studied an approach to generate promoters with higher expression activities and it was demonstrated only on 2 promoter examples (GPD and ZEV promoters). Also, the authors did not studied any changes of conserved motifs of these promoters. These limitations should be considered in the discussion.

Response to Editorial and Reviewer Comments

We would like to thank the editor and reviewers for their constructive comments and thoughtful suggestions. Below we address each of the comments provided.

Reviewer #1 (Remarks to the Author):

General Comments:

*This study focused on promoter engineering in the yeast *Saccharomyces cerevisiae*. The strong-constitutive GDP and orthogonal inducible pZEV promoters were randomised in spacer regions between known transcription factor binding sites TATA boxes, and transcription start sites. These promoters were used to control GFP expression, which was normalised to constitutively expressed mCherry in each cell to control for noise. Yeast populations transformed with diversified promoter libraries were then FACS sorted using 12 gates that spanned the range of GFP to mCherry values. Plasmid DNA was extracted from sorted populations and next-gen sequenced. Deep learning was then used to model the relationship between promoter sequence and GFP:mCherry expression. Predicted expression levels correlated well with actual measurements of those sequences.*

The model was then successfully used to predict novel sequences with higher expression levels than any sequence in the library or even the wild-type promoter sequences. Importantly, different methods of designing sequences were compared, with evolutionary optimisation found to give both high expression and high sequence diversity. This is important for future application of designed sequences as it minimised their homology and therefore loop-out potential when used multiple times in yeast. Another interesting finding was that the activation ratio of the inducible pZEV promoter could not be optimised because of an abundance of promoter sequences at the lower limit of detection in the training data. When individual promoters were tested alongside controls, the gradient ascent strategy was found to enable the design of a promoter with ~30% higher output than the native GPD promoter.

Finally, the importance of each base position was analysed alongside putative transcription factor binding sites and TATA boxes that may have been introduced in random sequences. Interestingly, new TF binding sites were significantly enriched in the high-expression sequences, as was a novel TATA box placement.

The study is interesting, thorough, novel, well-written, and of high fundamental importance. However, the results are underwhelming from an applied perspective given that only marginal improvements in expression of designed promoters could be achieved (or none except for leakiness in the case of pZEV) (Figure 4). My interpretation of this is that machine learning has limited potential to predict and engineer improved promoter designs, especially relative to other more common approaches. However, this could be due to the promoter library design, which left core promoter motifs unaltered.

I have no serious concern with the technical and scientific content of the article, but mainly question whether the results are impactful enough for Nature Communications. Regardless,

the comments below are made out of genuine curiosity, and might be useful for improving the manuscript.

Please note that I am not qualified to review the deep learning methodology. However, given the predictive power of the model and the empirical follow-up experiments I am unconcerned with this.

We thank the reviewer for the comments. In evaluating our results from an applied perspective, it should be noted that our model-guided design approach efficiently yields very large, sequence-diverse promoter sets which demonstrate high activity despite being much shorter than the benchmark sequences. Additionally, many of our design sets fulfil stringent limitations on GC content. These properties are all of practical importance for genetic engineers and synthetic biologists, but no previously reported promoter design approach addresses them all. Accordingly, we believe our results represent a significant advance beyond the current state of the art.

Major comment

1. Why is there such a large discrepancy between the improvement in GDP promoter expression seen in the designed promoters in Figure 3-a and that seen from the results in Figure 4-b? In 3a there is a doubling in expression level relative to the high expression-level training data while in Figure 4-b there is a ~30% improvement over the control promoter.

We thank the reviewer for the comment. All sequences in the training data have much lower activity than the control promoter tested in Fig. 4b, owing to the modifications made in library design (shortening the sequence and introducing variability). Bulk activity data for the pGPD library and for the pGPD benchmark sequence appear in Supplementary Fig. 2. Comparing the activities of the “Final” construct to the pGPD control shows that even the strongest training data sequences are indeed much weaker than the pGPD control.

In the methods section, the protocol used for measuring individual promoter activities (section 4.10) seems as though it may result in growth-stage variation between different strains. There is no control of inoculation density between the preculture and the main culture, which is not necessarily problematic, except that the main culture is grown for 24 hours prior to flow-cytometry measurement. Growth stage has a significant effect on promoter activity, and could explain the above discrepancy. Even if all cultures and replicates are in the same growth stage, it is uncertain whether they are in exponential growth, post diauxic shift, or stationary phase when the readings were taken. This could also explain why the TDH3 promoter has a 7-fold higher activity than TEF1 (Figure 4 b), which contradicts previous reports where pTEF1 activity is typically half of pTDH3.

We thank the reviewer for the insightful comment. It is indeed true that promoter activity is dependent on many variables, including culture conditions and growth stage. Inter-lab differences in culture conditions and protocols could lead to the different ratios of pTDH3/pTEF1 activity observed from group to group. To our knowledge, there is unfortunately no reliable protocol to ensure that these experiments are carried out under truly identical conditions; changing this would require not only fundamental metrology research but (perhaps

more difficult) agreement among the different groups working in this field as to which conditions to use.

Small growth stage effects may well explain some of the inherent biological noise in our measurements. However, it is unlikely that this affects the reliability of our results. Each sample measured on the flow cytometer arose from a separate colony of plasmid-transformed yeast (we have revised text in section 4.10 to clarify this point). Thus, growth stage noise would presumably be present from sample to sample, not just from strain to strain (which would be a more concerning situation with a risk of systematic bias).

FACS-seq measurements of designed promoters also provide a good control on growth phase, as the entire transformed library is grown in co-culture and is thus necessarily in the same growth phase when measured. We observed very good agreement ($R^2 = 0.92$) between designed sequences measured in FACS-seq and individually (Fig. 4a), supporting the conclusion that the level of growth stage noise in our measurements is acceptable.

Minor comments

2. How was diversity introduced to the promoter libraries during construction? The methods state that oligos corresponding to each variable region were PCR amplified and golden-gate assembled to make the promoter libraries. Was variation introduced solely through synthesis errors in the oligo pool? If so, then please make this clearer; if not, then please explain how genetic diversity was constructed. In general, some more detail on the library construction would be useful.

We thank the reviewer for the comment. For the pGPD and pZEV experiments, large portions of the oligos used in assembly were completely randomized, leading to an extremely diverse library. As mentioned in the main text, 83% of the pGPD library and 79% of the pZEV library sequence was randomized. We have added detail in sections 2.1 and 4.1 to clarify the library generation process.

3. I would expect that there would be increased GFP:mCherry variation in the promoter libraries relative to native promoter sequences. Did the distribution of GFP:mCherry values change in the diversified promoter libraries relative to the wt promoter sequence?

We thank the reviewer for the comment. Supplementary Figure 2 shows that the diversified promoter libraries indeed had a much broader distribution of GFP:mCherry ratios than the benchmark native promoter sequences.

4. During library construction, transformation, and outgrowth, were there any attempts to estimate the library size, or is the size estimate solely from sequencing?

We thank the reviewer for the comment. We did not perform NGS characterization of the library at earlier steps in the protocol to directly estimate its size and composition. However, our analysis of count distributions for read clusters (Supplementary Fig. 27) and unique sequences (Supplementary Fig. 29) provides some insight into this question. These results indicate that we

managed to characterize most of the sequences in our original libraries, although we would have benefited from greater sequencing depth in the 2x300 MiSeq runs had this been available. We were not able to consider this question in-depth in the manuscript due to space constraints.

5. Line 262 refers to a 'one-hot encoded dna sequence'. I thought this was a typo intended to be 'one-pot'. A brief explanation or contextualisation of 'one-hot' would make this section easier to digest for a broad readership.

We thank the reviewer for the comment. We have clarified the meaning of “one-hot encoding” in Section 2.2 in order to better present the work to a broad readership; we have also referenced the prior work which inspired this approach to provide additional context for interested readers.

6. What was the overall variant frequency for individual sequences in the promoter libraries?

We thank the reviewer for the comment. We could not go into great detail on this point in the main text for space reasons, but it is addressed by Supplementary Figures 27 and 29. We describe the results in terms of read counts instead of overall variant frequencies because the number of reads is most important for gauging whether we have adequate data to report a given sequence (e.g., sequences with only a single Miseq read were discarded (Supplementary Fig. 27)). We consider both variant frequencies in the final reconstructed sequences (Supplementary Fig. 27) and in the raw NGS data (Supplementary Fig. 29). As described in the text, the reconstruction strategy described in Sections 4.5 and 4.6 is necessary because at this length scale, individual reads will likely contain errors due to mutations in PCR amplification or even during sequencing-by-synthesis on the Illumina platform.

7. How was auto-fluorescence controlled? How were experiments conducted on different days normalised and compared?

Autofluorescence was controlled by comparison to a no-GFP control sequence (“Background” in Fig. 4e). mCherry was always driven by the strong pTEF1 promoter, so autofluorescence in this channel was quite low relative to the meaningful signal. We performed the manufacturer’s recommended calibration procedure before each flow cytometry run (this detail has been added to our Methods). Results from repeat measurements of the same sequences carried out on different days agreed well; accordingly, we performed no further normalizations in order to keep our data-processing workflow as simple and assumption-free as possible.

8. In the discussion it would be useful to contextualise and compare these results with other similar and recent efforts. For example, “Deciphering eukaryotic gene-regulatory logic with 100 million random promoters” published in Nature Biotechnology in December 2019.

We thank the reviewer for the comment. We have added a discussion of our work in the context of the recommended reference to the Discussion.

Reviewer #2 (Remarks to the Author):

This work is an interesting extension of the previously published research modeling yeast MPRA datasets of artificial 5' UTRs using CNNs (Genome Res. 27, 2015–2024 (2017) and Nature Biotechnology volume 37, 803–809 (2019) to analysis and modeling the whole promoter sequences. The article can be considered as an important evaluation of model-guided promoter design strategies and promoters' characteristics that are responsible for their activity. The authors characterized their models performance by R2 using 10% of the dataset for the test.

We thank the reviewer for the positive comments.

However, it might be that their training set had promoters highly similar by sequence with the test set examples and they predicted functional activity based mostly on sequence similarity (in such case the model will not work for promoters that provide high expression but very different by sequence from the training set). Ideally, they should get rid of highly similar sequences between their training and test set, then they can estimate the actual performance of their model in application to unseen sequences.

We thank the reviewer for the comment. We used highly degenerate libraries and an error-tolerant data analysis strategy (to prevent “different” sequences which are merely two versions of a parent sequence distinguished by a few mutations or sequencing errors) in this work. This strategy is described in Sections 4.5 and 4.6. Because of this, very few sequences in the dataset are at all similar to one another.

To further support this claim, we carried out a new analysis in which we randomly sampled 400 read clusters (each cluster is grouped as one final sequence as shown in Supplementary Figs. 26 and 27) for each dataset, and determined the distance to the “nearest” distinct sequence in the dataset, as well as the “farthest” distance between any two reads in the cluster. These results are described in Section 4.5 and presented as Supplementary Figure 28, and demonstrate that we successfully controlled the potential confound described by the reviewer.

CNN models for promoter identification in genomic sequences have been presented in PLoS One. 2017; 12(2): e0171410 and its following paper in Bioinformatics, 35 (16), 2730-2737 2019 describing possibility to reveal promoter functional sub-regions by in-silico mutation experiments. It would be interesting to see a discussion how the experimental technique applied in the paper increases the discovery power of pure computational approaches.

We thank the reviewer for recommending these interesting papers. We have added material to the Discussion section to contextualize and compare our approach with that adopted in these papers.

The authors claim (to provide a tool for generating promoters with user-specified functional properties) is an overstatement as in the article they just studied an approach to generate promoters with higher expression activities and it was demonstrated only on 2 promoter examples (GPD and ZEV promoters). Also, the authors did not studied any changes of conserved motifs of these promoters. These limitations should be considered in the discussion.

We thank the reviewer for these comments. The specific model we trained only predicts the sequence-function relationship for GPD and ZEV promoter sequences, and for the function of (uninduced and induced) promoter activity. However, our strategy of collecting a large dataset, training a model, and using the model to design sequences with properties the researcher specifies is likely to be more broadly applicable, so long as the property of interest is sufficiently amenable to high-throughput measurement for an accurate model to be trained.

We have revised our language regarding “user-specified functional properties” throughout for increased clarity, particularly to distinguish claims about the model and design strategies presented here from suggestions about the broader utility of our approach.

We further agree that the Discussion section ought to acknowledge and consider the fact that we did not test changes within the conserved motifs held constant by our library designs, and have revised it accordingly.

Reviewers' Comments:

Reviewer #1:

Remarks to the Author:

The authors addressed all of my original concerns and queries in their rebuttal and revised manuscript.

I am satisfied that the revised manuscript merits publication.

Reviewer #2:

Remarks to the Author:

The authors improved the manuscript during the revision. The paper describing significant results is ready for publication in Nature Communications.

The referees indicated that they were satisfied with our revisions to the original manuscript. Their comments appear below:

Reviewer #1 (Remarks to the Author):

The authors addressed all of my original concerns and queries in their rebuttal and revised manuscript.

I am satisfied that the revised manuscript merits publication.

Reviewer #2 (Remarks to the Author):

The authors improved the manuscript during the revision. The paper describing significant results is ready for publication in Nature Communications.